# Overview of Apoptosis, Autophagy, and Inflammatory Processes in *Toxoplasma gondii* Infected Cells

**DOI:** 10.3390/pathogens12020253

**Published:** 2023-02-04

**Authors:** Ehsan Ahmadpour, Farhad Babaie, Tohid Kazemi, Sirous Mehrani Moghaddam, Ata Moghimi, Ramin Hosseinzadeh, Veeranoot Nissapatorn, Abdol Sattar Pagheh

**Affiliations:** 1Drug Applied Research Center, Tabriz University of Medical Sciences, Tabriz 5166/15731, Iran; 2Infectious and Tropical Diseases Research Center, Tabriz University of Medical Sciences, Tabriz 5166/15731, Iran; 3Department of Immunology and Genetic, School of Medicine, Urmia University of Medical Sciences, Urmia 57147/83734, Iran; 4Cellular and Molecular Research Center, Cellular and Molecular Medicine Institute, Urmia University of Medical Sciences, Urmia 57147/83734, Iran; 5Immunology Research Center, Tabriz University of Medical Sciences, Tabriz 5166/15731, Iran; 6Department of Medical Immunology, School of Medicine, Tehran University of Medical Sciences, Tehran 14176/13151, Iran; 7School of Allied Health Sciences, Research Excellence Center for Innovation and Health Products (RECIHP), Walailak University, Nakhon Si Thammarat 81160, Thailand; 8Infectious Diseases Research Center, Birjand University of Medical Sciences, Birjand 97178/53577, Iran

**Keywords:** *Toxoplasma gondii*, apoptosis, necrosis, autophagy, inflammation

## Abstract

*Toxoplasma gondii* (*T. gondii*) is an obligate intracellular parasite. During the parasitic invasion, *T. gondii* creates a parasitophorous vacuole, which enables the modulation of cell functions, allowing its replication and host infection. It has effective strategies to escape the immune response and reach privileged immune sites and remain inactive in a controlled environment in tissue cysts. This current review presents the factors that affect host cells and the parasite, as well as changes in the immune system during host cell infection. The secretory organelles of *T. gondii* (dense granules, micronemes, and rhoptries) are responsible for these processes. They are involved with proteins secreted by micronemes and rhoptries (MIC, AMA, and RONs) that mediate the recognition and entry into host cells. Effector proteins (ROP and GRA) that modify the STAT signal or GTPases in immune cells determine their toxicity. Interference byhost autonomous cells during parasitic infection, gene expression, and production of microbicidal molecules such as reactive oxygen species (ROS) and nitric oxide (NO), result in the regulation of cell death. The high level of complexity in host cell mechanisms prevents cell death in its various pathways. Many of these abilities play an important role in escaping host immune responses, particularly by manipulating the expression of genes involved in apoptosis, necrosis, autophagy, and inflammation. Here we present recent works that define the mechanisms by which *T. gondii* interacts with these processes in infected host cells.

## 1. Introduction

*Toxoplasma gondii* (*T. gondii*) is an obligate intracellular pathogenic protozoanwith sexual and asexual life cycles that can infect any nucleated cell of most animals. The reason for the parasite’s success is partly due to its ability to manipulate host cells and escape immune mechanisms, resulting in chronic (lifelong) infections [1,2]. The life cycle of *T. gondii* is complex, with several infective forms and transmission pathways. Felines are definitive hosts of *T. gondii*, where the parasite reproduces sexually. The parasite undergoes sexual recombination in the cat and is then transmitted to intermediate hosts, including terrestrial and aquatic mammals and birds, as sporulated oocytes. However, new research shows evidence of sexual *T. gondii* in mice. Felines are the only mammals that lack delta-6-desaturase activity in their intestines, which is required for linoleic acid metabolism, resulting in systemic excess of linoleic acid. It was found that the inhibition of murine delta-6-desaturase and supplementation of their diet with linoleic acid allowed *T*. *gondii* sexual development in mice [3,4,5].

Ingestion of undercooked meat from intermediate hosts is also another source of transmission. After ingestion of sporulated oocytes or tissue cysts in contaminated meat, the parasite is released into the small intestine, where the infection process begins. Sporozoites or bradyzoites invade the host’s enterocytes, where they are thought to transform into fast-replicating tachyzoites [6,7]. Invasion of the host cells by the parasite results in the formation of the parasitophorous vacuole (PV), where the parasite resides (non-fusogenic) and escapes degradation from the endolysosomal system [8]. In a parasitophorous vacuole, the parasite can grow and replicate and, eventually, spread throughout the host [9].

In intact, healthy hosts, a strong induction of CD8^+^T cell-dependent immune responses occurs, which is essential for the control of acute (recently acquired) infections. Once the adaptive immune system is activated, the immunological pressure mounting on the parasite results in encystment in the muscle and, more commonly, in the brain [10,11]. Although primarily considered asymptomatic, *T. gondii* infection can result in different pathological conditions. These different disease outcomes, such ascongenital ocular toxoplasmosis, schizophrenia, epilepsy, or acquired toxoplasmosis, can be caused by different routes of transmission and different parasite strains that infect the host [12,13].

One process by which the parasite can manipulate the host is through regulated cell death (RCD), a fundamental process involved in cellular homeostasis [14]. RCD is a key processmediatedbetween the parasite and phagocytic cells, including neutrophils, macrophages, and dendritic cells (DCs) [15,16]. *T. gondii* can modulate the transcription of host cell genes, including those involved in modulating cellular metabolism and energy, immune responses, and signaling, that are involved in regulating RCD [2]. Moreover, during *T. gondii* infection, the parasite’s interaction with the host’s immune cells leads to the modulation of the host’s immune responses, especially through manipulating the expression of genes involved in apoptosis, necrosis, autophagy, and inflammation [17]. Collecting data on RCD in toxoplasmosis and studying biochemical and immunological pathways may help to clarify the parasite’s escape mechanisms and improve therapeutic approaches.

In this review article, we tried to clarify different aspects of the interaction of *T. gondii* with host cells concerning changes in apoptosis, necrosis, autophagy, and inflammation in the cell–parasite immune environment.

## 2. *T. gondii* Infection

There are three main lineages of *T. gondii* based on virulence factors determined in mouse infection models [18]. Type I strains (RH, CAST, GT1) have the most virulence, type II strains are moderately virulent (ME49, HART, WIL), and type III strains are considered avirulent (VEG, SOU, MOO) [19]. There are also many atypical genotypes in different parts of the world that exhibit characteristics of type I, II, and III strains [20]. The diversity of parasite strains triggers different responses in the immune system, leading to various biochemical changes and clinical diseases in the host [20].

The highest diversity of *T. gondii* genotypes is predominantly found in South America andAfrica [21,22]. Noteworthy, highly virulent strains can cause atypical infections in immunocompetent patients that may cause brain disorders such as schizophrenia. Combining the analysis of *T. gondii* strain types and the host-related factors (immune status and genetic background) may offer a better understanding of human susceptibility or resistance to *T. gondii* infection [23,24].

Animalsare mainly infected by the ingestion of infected hunts, congenital infections, or contact with oocytes. Humans commonly acquire *T. gondii* infection by ingestion of oocysts shed from the feces of infected cats [25,26]. Exposure to undercooked meat from infected animals, particularly pigs, is the main source of human infection in some countries, such as Poland where most pigs, cattle, and sheep are sources (around 80%) [19,27].Transmission of the parasite can also occur through ingestion of water containing oocytes, consumption of contaminated milk, and exposure to contaminated soil in yards and sandpits where children play and can get exposed [28]. Other important factors in the parasite’s transmission pathways include the ingestion of undercooked meat, especially rabbit [29], venison [30], raw oysters, mussels, and clams [31]; consumption of vegetables contaminated with oocytes and raw, unwashed fruits [32]; maternal-fetal passage of blood cells [33]; blood transfusions [34]; solid organ allografts [35]; allogeneic stem cell transplantation [36]; bone marrow dendritic cells (BMDCs) transplantation [37]; breast milk or breastfeeding [19]; sputum [38]; and semen [39] (Figure 1). The *toxoplasma* infection can be transmitted orally and vertically. If first contacted during pregnancy, *T. gondii* tachyzoites may pass to the fetus via the placenta. Reactivation remains the predominant route by which toxoplasmosis manifests especially among HIV/AIDS patients with a CD4 cell count less than 200 cells/μL. Lower socio-economic status, poor hygiene, and training or education may also contribute to high infection rates [40,41]. Dendritic cells (DCs) act as carriers of systemic parasites during infection [2,42]. It has been shown that the parasite can transmit from DCs to natural killer (NK) cells. Rapid transfer of *T. gondii* from infected DC to effector natural NK cells may contribute to the parasite’s sequestration and shielding from immune recognition shortly after infection [43,44].

The body’s response to the *T. gondii* challenge is to stimulate antigen-presenting cells, such as DCs and macrophages, and to activate cytotoxic T lymphocytes and release interferon (IFN)-γ, which is performed by inducible GTPases. IFN-γ-inducible effectors such as IFN-inducible GTPases, inducible nitric oxide synthase, and indoleamine-2,3-dioxygenase differentially play essential roles in the suppression of *T. gondii* growth [45].

## 3. Interaction of *T. gondii* with Immune Cell Signaling

*T. gondii* has highly specialized secretory organelles, which are involved in the host cell invasion [46]. Among the secretory factors, rhoptries (ROPs) and dense granules (GRAs) can be mentioned. ROPs secrete proteins that enable host-cell penetration and vacuole formation by the parasites, as well as evasion of the immune system [47]. Invasion is an active process that allows for the formation of the PV, which is considered a new organelle for the host cell [48]. During the invasion process and after PV formation, *T. gondii* secretes several specific parasitic proteins that disrupt several signaling pathways within host cells. These secreted factors provide a slight advantage to *T. gondii*, allowing it to evade immunity during theinitial invasion and after stimulating an effective immune response [49].

One of the important features of PV is the presence of the intravacuolar network (IVN) that connects the parasites to each other in the PV membrane. The IVN may be involved in virulence by helping route parasite rhoptry effectors released in the host cytosol back to the PV membrane, thereby preventing PV destruction by immunity-related GTPases [14,50].

An excellent example of parasite-specific host cell manipulation is via the *T. gondii* rhoptry protein 16 (ROP16). ROP16 is secreted into the host’s cell cytosol during the invasion and is stably combined with the signal transducer and activator of transcription STAT3 and STAT6 phosphorylation [51,52]. Phosphorylation of STAT6 by ROP16 leads to the induction of arginase-1. *T. gondii* is an arginine auxotroph; therefore, activation of arginase-1 by ROP16 depletes available arginine, an essential host-derived nutrient for the parasite [51]. Arginase-1 hydrolyzes L-arginine to produce urea and ornithine, which not only play a role in parasite starvation but also deprive macrophages of accessing substrate for NO production, an essential mechanism for intracellular pathogen destruction [53]. In addition, ROP16 can reduce the response of the immune system by stimulating the production of IL-4 and IL-10 [54].

Arginase-1 production is a characteristic of alternatively activated M2 macrophages via STAT3/6. M2 macrophages produce anti-inflammatory molecules that inhibit the T helper (Th) 1 response and may decrease host defense capacity [55]. M1 macrophages are essential for the early production of IL-12 and controlling of *T. gondii* infection [56]. A recent study has shown that *T. gondii* ROP16 is responsible for Stat3 activation and suppression of *T. gondii*-induced pro-inflammatory cytokines induced by type I strains [52,57]. Both STAT6 and STAT3 can suppress the production of IL-12 from macrophages. Mice infected with the ROP16-knockout parasites have shown that upon activation of ROP16, STAT3 inhibits Toll-like receptor (TLR) signaling and, ultimately, reduces inflammation [57,58,59].

A recent study by Johnson et al. has shown that type II *T. gondii* strain ROP16 leads to stable phosphorylation of STAT5, resulting in its translocation into the nucleus of the host cell, which is important for protective immunity in the gut mucosa of mice [60]. Schneider et al. demonstrated that STAT1 is activated in infected murine BMDCs (bone marrow-derived DCs) independent of serine 727 (Ser727) and tyrosine 70 (Tyr70) phosphorylation [61]. Despite its nuclear transcription, the tyrosine phosphorylated STAT1 could not dock to the *Irf1* gene promoter, as indicated by the lack of STAT1 complexes at the target site [62].

Immunological studies have revealed several parasite virulence factors regulated by the disengagement of immune activity. IFN-γ is essential for host cell resistance and acts by upregulating the expression of IFN-γ-activated effectors that destroy *T. gondii* [63]. Several parasite factors block the proper functioning of IFN-γ. One of these is the dephosphorylation of STAT1 by the suppressor of cytokine signaling 1 (SOCS1) [64]. Kim et al. showed that *T. gondii* infection downregulates the expression of SOCS1 in human fibroblasts. It is more likely that the block in IRF1 expression is due to a defect in STAT1 activation and/or its transcriptional activity in the nucleus [65]. *T. gondii* also prevents STAT1-mediated gene transcription by changing histone acetylation and other chromatin alterations of promoter regions where STAT1 binds [62]. *T. gondii* can also prevent the dissociation of STAT1 from DNA, which limits its ability to transcribe other STAT1-dependent genes [66].

SOCS3 and nuclear factor (NF)-κB are important host factors that regulate inflammation during acute *T. gondii* infection and that can also be manipulated by the parasite. Infection of knockout mice for the SOCS3 gene results in animal death. Mice with targeted deletion of SOCS3 in macrophages and neutrophils have reduced IL-12 responses and succumb to toxoplasmosis [67]. *T. gondii* upregulates the expression of SOCS2 on DCs by lipoxin A4 with anti-inflammatory activity and decreases the expression of IL-12 and C-C chemokine receptor 5 (CCR5) secretion [68]. Dense granular proteins 6 (GRA6) interfere with activated T-cell nuclear factor 4 (NFAT4), activating it via calcium-modulating ligands, which may lead to increased migration of inflammatory macrophages [69]. The release of GRA15 by type II strains, as opposed to types I/III, leads to host NF-κB activation and inhibiting IL-12 synthesis [70]. However, the *T. gondii* type I strain inhibits NF-κB activity via ROP18, leading to inhibition inthe expression of inflammatory cytokines and promoting parasite survival rate [71]. Host–parasite interactions via these parasite-specific factors result in life-long latent infection and increase the risk of transmission to a new host and the susceptibility in immunocompromised hosts [72].

## 4. Regulated Cell Death (RCD) and *T. gondii* Infection

RCD and the significant role of this process were described by Karl Vogt in 1842 [73]. RCD is required to regulate tissue growth and homeostasis. The term apoptosis did not appear until the end of the 20th century. RCD can occur either by apoptosis (also called programmed cell death) or necrosis [74].

Membrane-bound vesicles derived from apoptotic cells are now referred to as apoptotic bodies. Upon binding to specific receptors on macrophages, apoptotic bodies respond to death signals that induce the expression of anti-inflammatory cytokines, such as interleukins IL-10 and transforming growth factor (TGF)-β and IL-17 [75]. The morphological and ultrastructural characteristics of apoptosis consist of nuclear fragmentation, chromatin condensation, cell shrinkage, and the alteration of cell membranes, resulting in the formation of apoptotic bodies [76]. Another biochemical characteristic of apoptosis is the rapid phagocytosis of apoptotic cells by adjacent cells via receptors they express [74]. This is due to the translocation of phosphatidylserine from the lipid bilayer to the outer layers of the plasma membrane [77]. Phosphatidylserine and other cell surface factors such as calreticulin can act as ligands for phagocytosis receptors on phagocytes. It is noteworthy that cellular stress, including apoptosis, induces the expression of many stress proteins, including calreticulin; this may lead to increased amounts of calreticulin on the cell surface [78].

Host cell phosphatidylserine obtained by *T. gondii* can establish a balance between the parasite’s survival and the induction of the host immune response [79]. Guanosine triphosphatases (GTPases), such as immunity-related GTPase (IRGs) and guanylate-binding proteins (GBPs), are also important factors in the RCD process [80]. Binding and hydrolysis of GTP take place in the highly conserved G-domain, common to all GTPases [81]. Apoptosis is important for cellular homeostasis, and it can also be beneficial as a defense mechanism for host cells that encounter intracellular parasites [82]. Apoptosis of cells invaded by *T. gondii* may facilitate humoral immunity against the parasite [83], thereby promoting antibody responses and forcing intracellular pathogens to prevent parasite transmission in immature host cells. Intracellular pathogen-specific antibodies can bind to infected cells and thus mark them for destruction by Fc receptor (FcR)-bearing effector cells [84].

Necrotic cell death is also part of RCD but occurs via a different process. Necrosis is defined as the loss of cell membrane integrity by releasing nuclear and cytoplasmic contentsinto the extracellular space (Figure 2) [85]. The cellular morphology of necrosis is characterized by cytoplasmic vacuolation, cytoplasmic swelling (onychosis), and organelles (including the nucleus and mitochondria) [86]. These changes are due to ATP depletion and the inability of the membrane’s ion pumps to maintain the osmotic gradient stability [87]. Necrosis can also occur due to direct damage to the cell membrane, which eliminates cells without onychosis [88]. Plasma membrane disruption leads to the release of damage-associated molecular patterns (DAMPs), such as S100 proteins, heat shock proteins (HSPs), extracellular genomic, high-mobility group box 1 (HMGB1), heparan sulfate, monosodium urate (MSU), and mitochondrial DNA, which can stimulate inflammation [89]. The binding of a DAMP to its receptor activates an intrinsic inflammatory response and sends “endogenous adjuvant” signals that can stimulate DCs to promote T cell activation [90]. Necrosis of parasitized cells and the downstream induction of inflammation may be beneficial in activating neutrophils and macrophages to control *T. gondii* [91]. Specifically, macrophages play an important role in the pathogenesis of *T. gondii* [92]. They may be the preferred niche for *T. gondii* and, as a result, successfully promote host parasitism [93]. *T. gondii* can inhibit the development of immune responses against it by altering host gene expression, thus promoting the parasite’s escape and survival [59].

### Inhibition of Apoptosis by T. gondii

Growth and replication of *T. gondii* tachyzoites typically lyse infected cells after 72 h, depending on the type of organism [46]. Importantly, invading host cells do not undergo apoptosis during this period, but only die when the cells are lysed by parasites [71,94]. An important unanswered question is why and how the parasite prevents apoptosis. One possibility is through the development of modified host immune responses. The other is to keep the host cell alive so that it can provide essential nutrients, including cholesterol, purines, and amino acids, to the parasite. *T. gondii* can supply the required nutrients for growth, protection against host defenses, and replication and transmission [95,96,97]. Therefore, mechanisms that enhance the ability of the parasite to sequester nutrients appear to be a priority for the parasite to promote its survival. Moreover, an apoptotic cell is probably a poor supplier of critical constituents for parasite growth. By stopping apoptosis, the parasite ensures a stable source of metabolites for its growth. Inhibition of apoptosis, however, may also be critical for the chronic phase of infection. As bradyzoites grow slower in an infected cell, their priority changes from growth to conservation [98].

Studies have shown that intracellular parasitic infection leads to the inhibition of caspases, notably caspase 3, the bulge where signals converge along both pathways [82]. Considering the relevant factors, *T. gondii* infection simultaneously blocks the activation and activity of caspase-3. Despite the presence of caspase-3, other factors related to apoptosis were observed in *Toxoplasma-*infected cells, such as mitogen-activated protein (MAP) kinase (GRA24, ROP38), JAK/STAT (ROP16), NF-κB (GRA15), and P53 (GRA16). One of the underlying mechanisms by which *T. gondii* block caspase activity is through the secretion of the inhibitory apoptotic protein (IAP) in host cells, which effectively inhibits apoptosis [99]. There is persuasive evidence that *T. gondii* may inhibit caspase activity, triggering probiotic and anti-apoptotic responses in host cells [99]. The inhibitory effect of *T. gondii* on apoptosis can be controlled by regulating various signaling pathways, including the NF-κB and c-Jun N-terminal kinase (JNK) pathways [99,100].

Reactive oxygen species (ROS) can trigger trophoblast apoptosis by activating the JNK homologous protein, Anti-C/EBP (CHOP), and the caspase 12 pathway in *T. gondii* infection in mice [100]. It has been shown that distinct parasite strains elicited different responses in the host. Strikingly, *T. gondii* (type I-RH strain) modifies gene expression in mouse spleen cells. Gene expression was positively correlated with immune responses in the mice models [101].

Wang et al. reported that after treatment of Ana-1 cells with different concentrations of *T.gondii* excretory/secretory antigens (TgESAs), the proliferation and phagocytosis capacity of Ana-1 cells was decreased, and apoptosis was induced in a dose-dependent manner [102]. *T. gondii* inhibits caspase-3 and regulates the activation of different caspases upstream in the intrinsic or extrinsic pathway. Cells infected with *T. gondii* reduce levels of caspase-3, caspase-8, caspase-9, and caspase-12 [100,103]. A recent study showed that the caspase 12 inhibitor Z-ATAD-FMK (BioVision, Milpitas, CA, USA) effectively inhibited the activity of caspase-12 and caspase-3 in murine neural cells [104]. A variety of potential upstream changes were also observed, including overexpression of anti-apoptotic members of the Bcl-2 family proteins, initiation of the PI3-kinase signaling, and upregulation of STAT6. The activation of NF-κB by *T. gondii* is correlated with increased expression of anti-apoptotic genes such as the IAPs and Bcl-2 families [105].

*T. gondii* modifies the expression of apoptosis-related genes to maintain survival in host cells [71]. Nash et al. investigated the effect of numerous cell types infected with the RH strain of *T. gondii*. They found that intracellular tachyzoites of *T. gondii* were resistant to multiple inducers of apoptosis, including Fas-induced cytotoxicity, IL-2 deprivation, irradiation, UV, calcium ionophores, and beauvericin [106]. Granzyme B (released by NK cells and CD8^+^ T cells) acts directly on caspase and accelerates the induction of apoptosis. *T. gondii* reduces apoptosis in Granzyme B-dependent apoptosis in host lymphocytes [107,108]. *T. gondii* blocks host cell apoptosis by inhibiting caspases-3 and Granzyme B activity. The parasite-mediated inhibition of Granzyme B demonstrates that this inhibition is associated with the protection of infected host cells from cytotoxic lymphocyte-mediated apoptosis [109].

It has already been shown that *T. gondii* infection induces NF-κB activation. Therefore, the induced activation of NF-κB also enhances the activation of anti-apoptotic genes [71]. For example, it was shown that the role of NF-κB2 is the regulation of apoptosis after induction with the *T. gondii* ME49 strain. This strain upregulates A20, resulting in the inhibition of NF-κB activation and induction of apoptosis in human T cell lines [110,111]. Treatment with actinomycin D led to apoptosis in the cells. *T. gondii* infection inhibits actinomycin D in mouse spleen cells by inactivating caspases and activating NF-κB-mediated apoptosis [112].

Gavrilescu et al. showed that the activation of several apoptotic pathways induced by *T. gondii* depends on IL-12, p40, and FasL, which may play a role in the pathogenesis of fatal infections [113]. Nishikawa et al. showed that Fas/FasL interaction was involved in the apoptosis of fibroblasts from mice infected with the parasite, and the IFN-γ responsehad no significant effect on the initiation of apoptosis in *T. gondii*-infected cells [114].

It has been shown that NF-κB, which induces the anti-apoptotic pathway, induces the upregulation of Fas and FasL [115]. Treatment of CD8^+^ enriched splenocytes from immunized mice with concanamycin A, but not monoclonal anti-Fas ligand, significantly decreased their anti-proliferative and lethal abilities. Thisimplies that CD8^+^ T cells induced by immunization with RH antigen and live Beverley strain bradyzoites may exert protection against *T. gondii* infection, at least in part by granule-dependent cytotoxic activities [116].

It has also been suggested that Heat shock proteins (HSPs) are involved in the regulation of apoptosis in *T. gondii*-infected cells. HSP65 synthesis is inhibited via activation of the HSP65 antisense oligonucleotide by IFN-γ plus Tumor necrosis factor (TNF)-α. Hence, HSP65 appears to contributeto immune function by preventing apoptosis of infected macrophages and decreasing parasite survival and virulence. Under in vivo conditions, the Beverly strain, unlike the RH strain of *T. gondii*, induces the expression of HSP65 in the host gd-T cell [117].

Under in vitro conditions, the forced inhibition of HSP65 expression with HSP65 antisense oligonucleotides caused apoptosis during infection with the less contagious Beverly strain. Inhibition of apoptosis is only observed when HSP65 is expressed before cell infection, indicating that the timing of HSP expression may be crucial [82]. It is currently unknown how the expression of HSP65 interferes with the apoptotic pathway during *T. gondii* infection. One of the potential targets is an apoptosome because HSPs have been shown to interfere with the development of apoptosomes [118].

Inhibition of apoptosis is also transcriptionally regulated in favor of the parasite’s survival. Infection of mouse splenocytes activates host NF-κB transcription and anti-apoptotic genes [112]. Mitochondrial Mcl-1 is an essential signaling mediator that regulates autophagy and apoptosis in human mesenchymal stem cells (MSCs) infected with *T. gondii* [119]. After cell invasion, host cells increase the level of protein kinase B (Akt/PKB)/serine-threonine kinase and PI3K in a Gi-dependent manner, thereby delaying cell apoptosis [120]. In addition, *T. gondii* infection in vitro and in vivo induced the phosphorylation of PKB/Akt and Bcl-2-associated death promoter (Bad). Anti-apoptosis by *T. gondii* occurs partly through phosphorylative inactivation of Bad [121]. In the early stages of infection, *T. gondii* can cause apoptosis and lead to the spread of infection [14].

According to a previous study, the ROP18 protein of *T. gondii* causes host immunization and neuron apoptosis via endoplasmic reticulum (ER) stress [104]. ROP18 is released during the attack and placed on the membrane of the parasitophorous vacuole [122]. ROP18 has a considerable degree of homology with ROP2 [123,124], but has a high rate of polymorphism [125,126], and is regarded as the main determinant of the observed virulence differences between different strains [127].

In the murine model, ROP18 modifies innate and adaptive immune responses. ROP18 phosphorylates immunity-related GTPase (IRGs) and prevents the breakdown of the parasitophorous vacuole membrane (PVM) [128]. ROP18 has an arginine-rich region, and the disruption of this region has been shown to prevent the anchoring of ROP18 to the PVM; this results in the accumulation of IRGs in the PVM and their consequent destruction, indicating that ROP18 plays an essential role in the dysfunction of IRGs and that it depends on their proper localization [129].

Studies have also shown that *T. gondii* inhibits apoptosis by generating signals and activating transcriptional molecules. Serine protease, such as SERPIN B3/B4, is extensively expressed in *T. gondii*-infected macrophages upon STAT6 activation. In *T. gondii*-infected macrophages, upregulation of the squamous cell carcinoma antigen 1/2 (SCCA1/2) gene by the host transcription factor STAT6 may be an important mechanism for *T. gondii-*mediated apoptosis in host cells [130]. The STAT3-miR-17-92-Bim pathway provides a mechanism for inhibiting the host cell apoptosis after *T. gondii* infection. In other words, STAT3 mediates a prosurvival pathway by upregulating the *miR-17–92* miRNAs that in turn targets Bim, leading to the survival of host cells with *Toxoplasma* infection [131].

## 5. Autophagy in *T. gondii* Infection

Experimental evidence suggests that autophagy can kill many pathogens such as *T. gondii*, particularly in murine models. In this regard, it is important to produce IFN-γ in the early phases of infection [132]. Influenced by IFN-γ, infected host cells react by regulating about 2000 genes, called IFN-induced genes. These transitional molecules, such as IRGs and guanylate binding protein (GBP), rapidly accumulate around the PVM, leading to the disruption of the PVM and parasite death in murine cells [133]. In a mouse model, it was revealed that Autophagy-related (Atg) 5 and Atg8 are vital for correctly targeting *T. gondii* PVM [134]. In addition, Atg3, Atg7, Atg12, and Atg16L1 are recruited into the PVM to stimulate parasite death (Figure 3). In infections with viral strains of *T. gondii* (e.g., type I), it has been shown that the parasite-secreted antigens, such as ROP5, ROP17, and ROP18, are involved in maintaining the integrity of the PVM. The ROP18 kinase function begins after initiating the IFN-γ cascade and protects the degradation of PVM by blocking IRG-dependent killing through the inactivation of IRG proteins by phosphorylation of the nucleotide site [128,135]. This effect involves simultaneous expression of a ROP5 allele that can be accessed with ROP18 at the Irg phosphorylation site [136].

*T. gondii* can also be killed independently of IFN-γ by autophagy in mouse macrophages by the involvement of the TNF receptor superfamily, AMP-activated kinase, calcium/calmodulin-dependent kinase β (CaMKKβ), activation of JNK, and Unc-51-like autophagy activating kinase 1 (ULK1) [64].

In murine cells, STAT1 signaling initiates the expression of inducible nitric oxide synthase (iNOS), generating ROS and nitric oxide (NO), as well as the upregulation of GBP and IRG. The recruitment of GBP and IRG in PV depends on the main set of autophagy proteins, containing the Atg5-12-16 complex. Suwanti et al. demonstrated that murine congenital toxoplasmosis increases skull apoptotic index and skull apoptosis associated with increased IFN-γ expression, but it decreases TNF-α expression [137]. Blockade of host apoptosis by Z-VAD-FMK can reduce TNF-α-induced output, whereas blockade of necroptosis by necrostatin-1 has a limited impact on induced TNF-α [138]. Among the identified proteins, the immunity-related proteins N-myc and STAT interactor, IL-20RB, IL-21, ubiquitin C, vimentin, and the apoptosis-related protein P2RX1 were further verified as ROP18Itargets using sensitized emission-fluorescence resonance energy transfer (SE-FRET) and co-immunoprecipitation [139].

### 5.1. Host Cell Autophagy Pathways Targeting T. gondii

*T. gondii* provides an excellent model for determining whether the immune system can target pathogens to lysosomal degradation. CD40 is a major regulator of cell-mediated immunity, which kills *T. gondii* by macrophages and requires the recruitment of autophagosomes around the PVs, leading to lysosomal degradation of the parasite [140].

According to Choi and colleagues (2014), IFN-γ can increase the degradation of PV and parasite antigens; hence, it can be identified by *T. gondii-*specific T cells [134]. Recent studies in human and murine cells have shown that virulent strains of *T. gondii* can inhibit the function of IFN-γ by blocking IRG and IFN-γ activation and stimulating the formation of PVM and destroying PVP [141,142,143]. However, all recognized strains prevent destruction by IFN-γ if they infect host cells before being activated with IFN-γ. This is because the parasite deregulates gene expression induced by IFN-γ, such as blockingGBP, IRG, iNOS, indoleamine-2, 3-dioxygenase (IDO), and Major histocompatibility complex (MHC) classes I and II [65,144]. In HeLa cells stimulated with IFN-γ, ubiquitin is located around *T. gondii* type II and III PVS to surround and destroy them through the process of autophagy. This pathway uses the human-specific nuclear domain protein (NDP) 52 and p62-binding proteins and is dependent on Atg16L1 and Atg7 [145]. In contrast, in HAP1 cells, the Atg16L1 knockout does not affect the type II limitation of *T. gondii* and has only a marginal effect on the recruitment of GBP. In human foreskin fibroblasts (HFFs), the destruction of Atg5 did not affect the IFN-γ-mediated growth of *T. gondii* type I. Moreover, in human umbilical vein endothelial cells (HUVECs), autophagy and deletion of *T. gondii* type II were observed. Atg9 is required for the survival of *T. gondii* in immune cells, and the general parasite virulence may be promoted by interference with a canonical pathway of autophagy [146].

In many types of human cells, *T. gondii* IFN-γ-mediated restriction is mediated by IDO upregulation, which degrades L-tryptophan and inhibits the progression of auxotrophic tryptophan. In human cells, upregulation of IDO and ROS may lead to the control of parasite proliferation. The activity of inorganic nanoparticles (NPs) may be due to alterations in the redox condition and potential of the parasite’s mitochondrial membrane [47].

Adeyemiet al. showed the interaction between several host cellular processes, including Hypoxia-inducible factor 1-alpha (HIF-1α) activity, IDO activity, and, more broadly, the tryptophan pathway, contribute to theanti-parasitic effect of NPs [147]. Therefore, these pathways are probably to oppose and coexist. Nutrient deficiencies can lead to autophagy which can deliver important proteins for GBP and autophagy function (e.g., LC3 and ubiquitin) to autophagosomal rather than vacuolar membranes. How exactly *Toxoplasma* is restricted in a human cell might depend on the phagocytic ability of the cell versus the induced GBP and IDO levels. Surprisingly, GBP versus IDO-mediated restriction of *Toxoplasma* has not been studied. It is thus possible that these pathways counter-regulate each other and co-exist [134].

The molecular mechanism of PVM detection by GTPases is still an open question. Studies show that the C-terminal isoprenylation of GBP2 regulates GBP2 use in PVM. However, isoprenylation alone cannot differentiate PVs from host organelles. The detection of ubiquitin in intracellular organisms creates an important host mechanism [45]. More types of ubiquitination in *T. gondii* PVM are known as linear M1, K48, and K63 polyubiquitin chains [148,149]. TNF receptor-associated factor 6 (TRAF6) and Tripartite motif-containing protein 21 (TRIM21) play an important role in regulating ubiquitination in *T. gondii* PVM after IFN-γ treatment. However, the effect of TRAF6 in stimulating IFN-γ to suppress *T. gondii* is controversial [145,148]. PVM disruptions and ubiquitin accumulation in PVM were largely unaltered in TRIM21-deficient cells, cy, suggesting that compensation of ubiquitination in *T. gondii* PVM can be achieved by other E3 ligases, such as TRAF6 [150].

Atg8 is aubiquitin-like protein, essential for the formation of autophagosome membranes. In mammals, Atg8 includes the subclass of LC3, such as A, B, and C types, and GABARAP subunits, including Gabarap, Gabarapl1, and Gabarapl2/Gate-16 [151]. Studies have revealed that an increase in the accumulation of LC3 in PVM *T. gondii* could be a result of cell activation by CD40. Stimulation of CD40 may lead to vacuolar ATPase activation followed by fusion of pjosphoinositide-3-class 3 (PIK3C3) and Rab7 on PVM with late lysosomes or endosomes [152]. However, the recruitment of LC3 in *T. gondii* PVM is controversial in response to IFN-γ activities [153]. It is possible that autophagy is not functionally involved in IFN-γ mediated anti-parasitic immune responses. In autophagy, Atg5–Atg12 combines with the phosphatidylethanolamine (PE)-Atg8 conjugation reaction, distinctively promoting protein–lipid conjugation [154]. The expression of Atg5 in phagocytic cells is essential for cellular immunity against intracellular pathogens, and autophagic proteins may participate in the elimination of immunity and intracellular destruction of pathogens by the autophagosomal processes. Members of the Atg3 family belonging to the Atg12 integration system, including Atg3 and Atg7, also act as intermediates to produce IFN-γ against *T. gondii*. However, other ATG proteins that regulate the pathway contain Atg9, ULK1, and PI3K. Therefore, anti-IFN-γ-mediated *T. gondii* responses can be regulated by these Atg proteins independently of autophagy [155,156,157]. The GABARAP subfamily member of the Atg8 family was shown to critically control the uniform localization of the GBP [158]. GBP is located in vesicle-like structures and is also distributed in IFN-γ-producing cells. In toxoplasmosis, the growth of GBP in host cells reduces the genetic barriers involved in autophagy [159]. Conversely, the uptake and proliferation of fatty acids (FAs) by *T. gondii* are increased in host cells without mitochondrial fusion, required for efficient mitochondrial oxidation of FA, or when mitochondrial oxidation of FA is pharmacologically inhibited [160]. The association between LC3 and the PV membrane results in the direct transplantation of lipidated LC3 into the membrane and not through the fusion of LC3-positive autophagosomes with PV [161]. LC3 is a single membrane phagosome containing extracellular pathogens or dead cell debris, in contrast to canonical autophagy, in which canonical LC3 is recruited to double-membrane autophagosomes [162,163,164].

### 5.2. Mechanisms of Autophagy Control in T. gondiiInfection

Understanding the mechanisms of autophagy in *T. gondii* may pave the way for disease control. GTPases such as GRI and GBP are two families of proteins that have evolved as effective mechanisms for controlling anti-*T. gondii* autophagy pathways. The homeostasis of these two families is controlled by a set of autophagy proteins involved in the prolonged phase of autophagosome formation. Removing any of these proteins leads to the natural activation and combination of IRGs and the disruption of *T. gondii* infection control [158,165].

In addition, because *T. gondii* is a tryptophan auxotroph, tryptophan degradation and indoleamine oxidase induction are important for parasite control in specific cell types [166]. Muniz-Felicianoet al. showed that, in cells that were not subjected to immune or pharmacologic upregulation of autophagy, blockade of the epidermal growth factor receptor (EGFR) resulted in parasite encasing by structures that expressed the autophagy protein LC3, vacuole-lysosomal fusion, and autophagy protein-dependent killing of the parasite [167]. Portillo et al. showed that the killing of *T. gondii* could be inhibited by the expression of the dominant negative protein kinase R(PKR). Thus, *T. gondii* activates a Focal adhesion kinase (FAK)→ Src → Y845-EGFR → STAT3 signaling axis within mammalian cells, which allows the parasite to survive by avoiding autophagy through a mechanism that may block the activation of PKR and the Eukaryotic Initiation Factor 2 alpha (eIF2α) [168]. In murine cells, STAT1 signaling initiates inducible nitric oxide synthase (iNOS) expression, generating ROS and NO, as well as positive regulation of GBP and IRG. Studies have shown that tyrosine kinase inhibitors and EGFR can inhibit the phagolysosomes in parasites. EGFR activation depends on the expression of at least two microneme proteins, MIC1 and MIC3. *T. gondii* micronemal proteins containing EGF domains appeared to promote EGFR activation [167].

In addition to plasma membrane receptors, intracellular proteins may also be targeted during *T. gondii* infection due to the presence of the rhoptry protein [169]. Previously, it was indicated that rhoptry proteins only affect the infected cell when injected concurrently with the invasion [170]. This paradigm is also challenged by a system in which *T. gondii* is designed to secrete Cre recombinase into host cells. Cre interrupts a stop codon, and then GFP can be identified in infected and uninfected cells, respectively [171]. Compared to the plasmid encoding enhanced green fluorescent protein (pEGFP), pEGFP-GRA15II transfection facilitated cell apoptosis, increased expression of caspase-3, caspase-4, homologous protein C/EBP (CHOP), and binding protein-1 X-box (XBP1), a 78-kDa glucose-regulated protein (GRP78), induced ER stress and, subsequently, caused apoptosis of choriocarcinoma JEG-3 cells [172]. In addition, the nuclear translocation of pSTAT6, which depends on ROP16, can be detected in vivo and in vitro in uninfected cells, which agrees with findings that rhoptry proteins enter these uninfected cells [173]. Importantly, rhoptry excretion in non-infectious cells is apparently a widespread phenomenon and can be created in various immune and non-immune cells. The interaction between the host and the pathogen is a continuous struggle, and we suggest that modulating *T. gondii* microenvironment offers two important advantages to fighting off this parasite [174].

The first activation of the host cellular process during the parasitic invasion gives the parasite opportunity to carry out the propagation process. This involves altering host-cell metabolism to help parasites access essential nutrients and activating mechanisms to prevent inherent immune defense such as apoptosis, pyroptosis, and autophagy. Second, it would allow the parasite to disarm IFN-γ and other mechanisms of destruction of immune effectors before entering the host cell. The ability to access and regulate immune responses may provide another mechanism for the parasite to escape the immune response. For example, activation of HIF-1 may promote the development of effector and regulatory T cells [175]. Downregulation of T cells not only aids in immune evasion but may also affect immunodeficiency-related complications. Cellular immunity mediated by T cells is essential to resist primary infection and for maintenance of quiescence during latent *Toxoplasma* infection [176].

## 6. The Role of RCD in Controlling *T. gondii* Infection

CD40 is expressed on the surface of antigen-presenting cells (APCs), such as macrophages and many non-hemopoietic cells. In this case, these agents activate pro-inflammatory mediators, such as IL-12, and ultimately lead to IFN-γ secretion. In human cells, CD40 is associated with its ligands expressed on the surface of T cells and leads to the production of IFN-γ from T cells, dependingon the production of IL-12 from macrophages [177,178].

The interactions of CD40-CD40L result in the production of TNF-α required for CD40-CD40L signaling to eliminate *T. gondii* containing PVs by combining LC3 and lysosome proteins. Ubiquitination in human cells and recruitment of automated adapters are not required in the presence of GBP. The recruitment of IRGs and GBP for PVM depends on the Atgs. HeLa epithelial cells target ubiquitinated PVs to improve retardation in cells through non-canonical autophagy. This host defense pathway and the one described in endothelial cells are avoided by type I parasites, suggesting the presence of a human *T. gondii* virulence factor [155].

In HFFs, the destruction of ATG5 does not affect the IFN-γ-mediated growth restriction of *T. gondii* type I. In other words, *Toxoplasma* resistance is not significantly altered in cells deficient in ATG5 [179]. In human vascular endothelial cells, p62-dependent endo-lysosomal acidification, independent of autophagy and ubiquitin and *T. gondii* type II clearance has been observed [149]. The role of Atg proteins in controlling *T. gondii* in human macrophages has not been established. Autophagy is a pathway closely linked to cellular metabolism. In many cell types, the IFN-γ-mediated limitation of *T. gondii* is implemented by the overregulation of IDO, which, by mortifying L-tryptophan, prevents the evolution of auxotrophic tryptophan in *T. gondii* [180].

NACHT leucine-rich-repeat protein 1 (NALP1) alleles produce pyroptosis in human cells during *T. gondii* infection. Similarly, the gene Arachidonate 12-lipoxygenase, type 12S (*ALOX12*), which encodes the arachidonate 12-lipoxygenase enzyme, has alleles related to toxoplasmosis, and *ALOX12* knockdown prevents the proliferation of *T. gondii* tachyzoites by aggregating RCD [181]. An important example of resistance to cell death is the purine receptors (P2X7R) on macrophages that are activated by extracellular ATP and can be upregulated by IFN-γ and TNF-α. Human polymorphism in the P2X7R gene may affect susceptibility to *T. gondii* infection in people with the loss of P2X7R function [182].

RCD prevents *T. gondii* proliferation, which requires replication and survival of the host cell. Therefore, it is not astounding that *T. gondii* inhibits host cells’ apoptosis; this inhibition can occur via various pathways (Figure 4). Inhibition of proteolytic proenzymes, such as caspases 3/7/8, can trigger RCD upon activation [175]. However, the inhibition of caspase-8 sensitizes cells to necroptosis, because caspase-8 is an inhibitor of serine/threonine protein kinase (RIPK3), a key mediator of necroptosis [183]. Therefore, it is likely that when IFN-γ, TNF-α, or TLR3 are activated, they can all activate RIPK3; hence, *T. gondii*-infected cells can be destroyed by necroptosis. HFFs, generally, accelerate the growth of *T. gondii*. However, IFN-γstimulated HFFs by an unknown mechanism when infected with a type I *T. gondii* strain, causing an early onset of the parasite [179].

It has been indicated that human cytotoxic T cells can kill *T. gondii*-infected cells through pore-forming perforins and secrete granzymes through these pores, which kill infected cells by triggering caspases. Human T cells also secrete antimicrobial guanidine peptides via these pores, which are capable of destroying the *T. gondii* PVM, so that the granzymes enter and kill the parasite by producing ROS [107]. Lewis rat macrophages are naturally resistant to *T. gondii* infection, probably because they undergo rapid NLRP1-facilitated pyroptosis during infection, thereby eliminating their reproduction niche. In humans, the NALP1 gene is associated with susceptibility to congenital toxoplasmosis. The destruction of NALP1 in a single human cell leads to decreased levels of IL-1β, IL-18, and IL-12, increased parasite proliferation, and acceleration of monocyte death [184].

## 7. Inflammation-Associated Factors in Toxoplasmosis

The inflammatory process involves the assembly process in the cytoplasm following the detection of proteins in the environmental and microbial hazard signal complex [185,186,187]. The inflammation process is regulated by the GBP in response to *T. gondii* infection in the host cells. The active caspase-1 then cleaves pro-inflammatory cytokines pro-IL-1β and pro-IL-18. The NLRP 1, NLRP 3, and NLRP12 inflammasome are important for the control of *T. gondii* in mice [188,189,190] and rat macrophages, respectively [189]. The NLRP 1 and NLRP 3 are significant for the in vivo control of *T. gondii* proliferation. Innate resistance to acute toxoplasmosis is dependent on the activation of both TLR and NLR sensors [189]. NlRC4, NLRP6, NLRP8, NLRP13, AIM2, and neuronal inhibition of apoptosis inhibitory protein (IPAN) increase in THP-1 cell lines, and the role of inflammation may be critical to the response to *T. gondii* infection [191,192].

Rapid cell death has been observed during the invasion of IFN-γ-stimulated fibroblasts in mice and humans by *T. gondii* [153,179]. Cell death in IFN-γ-stimulated fibroblasts during infection with type II and III strains of *T. gondii* was associated with degradation of the IRG-based PV membrane and did not reduce apoptosis. However, pyroptosis can be triggered without cleavage of caspase-1 [153]. Activation of IFN-γ in infected cells can operate on p65 guanylate binding proteins or GBPs and IRG or G47-GTPases immunogenic, capable of destroying otherwise invulnerable PV [193,194]. Studies have shown that cells infected by virulent strains of the parasite rarely undergo necrosis. According to Zhao et al., the autophagy process does not play a significant role in the main stages that lead to parasite death. They concluded that IRG protects *T. gondii* infection through a new mechanism involving the decomposition of the vacuolar membrane and, ultimately, leading to necrosis of the infected cells [153]. Studies have shown that given the absence of TLR11 or TLR12 expression in humans, monocytes in response to infection by *T. gondii* produce inflammatory cytokines in humans, suggesting that other TLRs in humans recognize different behavior of *T. gondii* for the production of IL-12 in infected human cells [195].

Immunosuppression after activation is also an important step in preventing immunopathology during *T. gondii* infection. IL-10 and IL-22 are members of the IL-10 family of cytokines, which have anti-inflammatory effects [196]. In addition, parasite growth can increase when the host cell viability and the expression of IL-1 and IL-18 are reduced in monocytes modified to express reduced levels of NLRP1 protein [197,198].

The contribution of interactions between human toxoplasmosis and the inflammatory process indicates that the expression of IL-1 in human monocytes depends on ASC and caspase-1, which is an adapter protein that binds caspase-1 to NLRP3/CARDB or NLRP1/caspase-5 in the flammomasome [198]. IL-1 expression was significantly increased by the type II parasite strain (Pru strain), and these effects depend on GRA15, which increases IL-1 expression by activating NF-κB [198]. ROP16 kinase has been shown to phosphorylate STAT3/6, which suppresses the phosphorylation of NF-κB and thus reduces inflammation [52,57]. Studies have shown that the human monocytes infected with *T. gondii* types I, II, or III do not cause rapid cell death [197,198]. Therefore, in human monocytes, the activation of the inflammation process limits parasite growth by mechanisms dependent on IL-1 and IL-18, or with some other unknown mechanism. This method of toxoplasmosis control is distinct from that observed in macrophage-resistant *T. gondii*, in which the activation of NLRP1 inflammation leads to rapid cell death. Strains of mice that are susceptible to *T. gondii* confer a difference from a 1.7-cM genetic source, the toxol locus [199]. In susceptible mice, macrophages infected with *T. gondii* do not undergo pyroptosis and do not secrete IL-1 [189], whereas macrophages secrete IL-1 and undergo pyroptosis in resistant mice [188]. This difference causes inflammation and then leads to a difference in the spread of parasites in macrophages [189]. In mice, NLRP1 and NLRP3 contribute to inflammasome activation with *T. gondii*, as shown by the induction of IL-1 and its secretion by macrophages. NLRP1 and NLRP3 are innate immune sensors for *Toxoplasma* infection, activated via a novel mechanism that corresponds to a host-protective innate immune response to the parasite [188,189]. Contrary to results obtained from rats and similar to those observed in humans, inflammasome activation is not the underlying cause of pyroptosis [188]. Several specific combinations have been found to trigger inflammation in mice when macrophages encountered substances, such as Pam3CSK4 and lipopolysaccharide (LPS) [188,189]. The influence of *T. gondii* on activating pyroptosis/inflammation highlights the importance of the host cell life for parasite survival.

## 8. Effect of *T. gondii* on Autoimmune Diseases

Autoimmune diseases can be caused by different factors, with genetics and environment being the most important factors, and the effective mechanisms are molecular mimicry and superantigens [200]. Some parasites can worsen and sometimes improve disease symptoms with different effects, one of which is *T. gondii*. Various evidence has shown that *T. gondii* infection plays a role in thyroid autoimmunity and rheumatoid arthritis using severalmechanisms [201]. As mentioned before, this parasite may play an important role in activating several types of autoimmune diseases (AD_S_) by using HSPGS and SAG1 as receptors, infecting various cell types [202]. We can point out the effects of various vitamins and ionic substances that are affected by this parasite and play an important role in our immune system, such as iron and folic acid, and vitamins such as vitamin D [202,203]. The effect of cytotoxic lymphocytes and the secretion of anti-inflammatory cytokines such asIL-4 and IL-13 due to this disease and their role in autoimmune diseases can also be considered [204]. As mentioned above, the effect of factors such as apoptosis and autophagy, which were aforementioned in detail, and the role they play in the body and cell homeostasis, can be considered one of the most important factors in developing autoimmune diseases [205,206].

## 9. Conclusions

A better understanding of the cellular processes by which *T. gondii* immuneregulates host cell death will boost our comprehension of the host–parasite interplay in toxoplasmosis. The various secretory organelles of *T. gondii* are involved in the invasion and modulation of host cell functions. In general, a set of RCD processes is used by the parasite for this purpose. The recognition, contact, and adhesion to the host cell seem to be mediated mainly by SAG and MIC proteins. Different cell lines of the same species can vary significantly in the expression of RIPK3, NLR, and IRG, which probably explains the cell type and species differences in response to *T. gondii*. Critical to clarifying the in vitro and in vivo role of RCD blockade may be the identification of defective parasite mutants and the modulation of cell signaling pathways. Thus, the parasite achieves an adequate balance between the RCD response and its evasion to achieve maximum dissemination in the body. Pharmacological approaches to boost autophagy for therapeutic aims may be intricate by the potential role of autophagy in several cellular functions, the intricacy of autophagy pathways, and the specificity of pharmacological agents. Strategies to prevent *T. gondii* from inhibiting autophagic targeting may represent a novel strategy to improve the treatment for toxoplasmosis in immunocompetent individuals and reactivated toxoplasmosis in immunosuppressed individuals. The description of the molecular processes by which different strains of *T. gondii* manipulate autophagy and cell death pathways may also serve as an advantage to investigate new pathways in normal cells and in diseases in which autophagy and other RCD processes play a key role.

## Figures and Tables

**Figure 1 pathogens-12-00253-f001:**
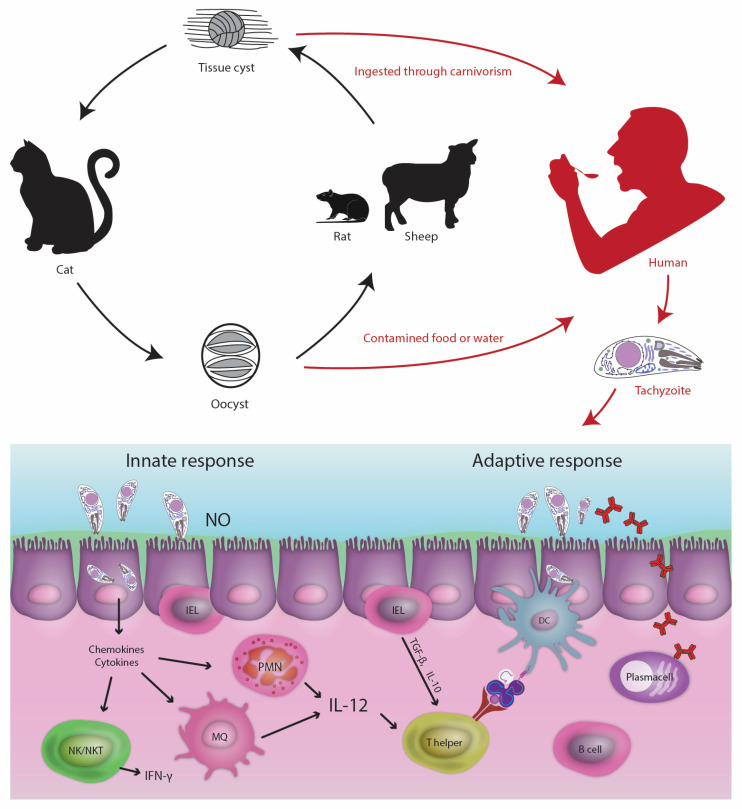
Life cycle of *T. gondii*. Schematic representation of the three virulence stages, main infection routes, and host innate and adaptive immune responses to toxoplasmosis.

**Figure 2 pathogens-12-00253-f002:**
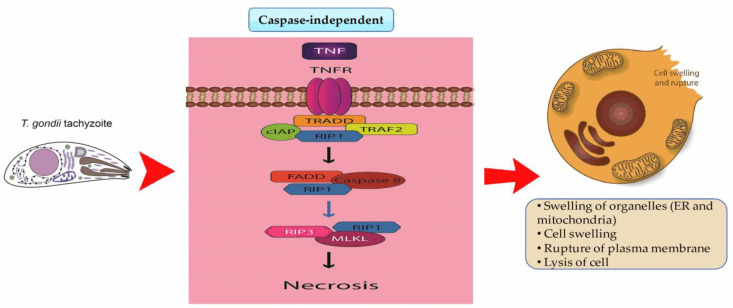
Necrosis occurs in caspase-independent processes. TNFR recruits TRADD; this recruitment allows the formation of different complexes related to the RIPK1 protein or pro-caspase 8. TRADD-FADD-pro-caspase-8 allows caspase-8 activation. Necrosis is dependent on RIP1, which is the target protein in necrotic cell death. The RIP1/RIP3 necrosome forms a functional signaling complex required for programmed necrosis. Given these factors, necrosis occurs due to the loss of cell membrane integrity by releasing nuclear and cytoplasmic contents into the extracellular space.

**Figure 3 pathogens-12-00253-f003:**
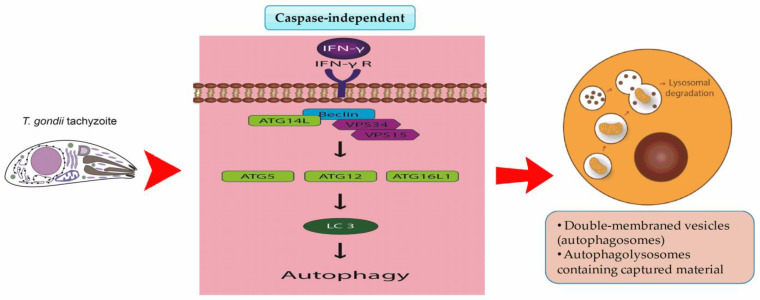
Autophagic cell death is a caspase-independent process. IFN-γ induces the degradation of the parasitophorous vacuole (PV) and parasite antigens. The activity of VSP34, which binds to beclin 1, requires the activity of VSP15, the beclin 1 regulator, and ATG14L. Atg5-Atg12/Atg16L1 targets cytosolic LC3 (LC3 I) to the isolation membrane, where it turns into LC3 II by conjugation with phosphatidylethanolamine, an effect driven by the E3-like enzymatic activity of Atg5–Atg12. *T. gondii* infection leads to the generation of large LC3-positive structures surrounding the PV and increased levels of active and lipidated LC3, which may result in the enhanced flow of autophagic degradation products into the vacuole. *T. gondii* in the cytoplasm that subvert phagolysosomal degradation usually result in the start of autophagy and are cleared via digestion in the autophagolysosome/authophagosomes process.

**Figure 4 pathogens-12-00253-f004:**
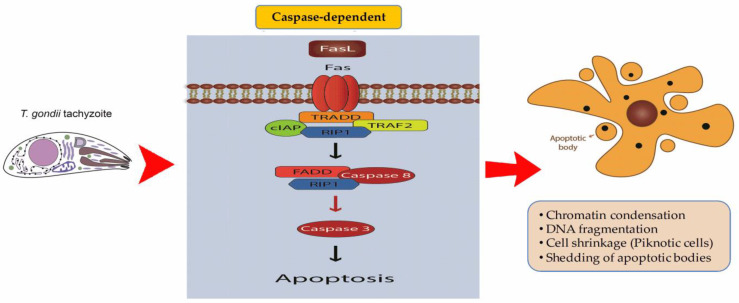
Apoptosis Pathways; The binding of TNF-α to its receptor, TNFR1, induces the assembly of TRADD, Rip1, and TRAF2 in complex I of the surface membrane. Complex I is then released into the cytoplasm, where FADD may attach to form complex II, which acts as a scaffold for caspase-8 binding and activation. FADD is a 28-kDa adaptor protein that is a critical component of the death receptor apoptotic signaling pathway. Caspase-8 has been shown to cleave and inactivate RIP1 during apoptosis. The ultrastructural characteristics of apoptosis consist of nuclear fragmentation, chromatin condensation, cell shrinkage, and the change of cell membranes, resulting in the formation of apoptotic bodies. Different mechanisms of *T. gondii* contribute to the inhibition of host cell apoptosis during infection. Cells infected with *T. gondii* reduce the levels of caspase-3, caspase-8, caspase-9, and caspase-12. *T. gondii* inhibits caspase 3 and regulates the activation of different upstream caspases in the intrinsic or extrinsic pathway.

## Data Availability

Data that support the findings of this study are available from the corresponding author upon reasonable request.

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
