# Peer review of "Overview of Apoptosis, Autophagy, and Inflammatory Processes in Toxoplasma gondii Infected Cells"

_pathogens, 2023, doi:10.3390/pathogens12020253_

Round 1

Reviewer 1 Report (Previous Reviewer 1)

Dear Editor, the manuscript is very nice and  improved alot and I propose to accept the current version.

Author Response

We would like to thank you very much for the very thoughtful review that helped us greatly to revise the manuscript and to improve its quality. 

Reviewer 2 Report (Previous Reviewer 2)

The authors have greatly improved the pharsing and grammar of the article, which was my most critical point. They have also added an additional figure do depict the lifecycle of the parasite Toxoplasma gondii, which greatly improves the understanding of the text.

Author Response

We would like to thank you very much for the very thoughtful review that helped us greatly to revise the manuscript and to improve its quality. According to your constructive comment, the manuscript has been studied several times by the authors, and its grammatical and structural problems were corrected. In addition, the manuscript was completely revised by English editors.

Reviewer 3 Report (New Reviewer)

Dear Author,

This is a straightforward review reporting overview of the immune response against Toxoplasma gondii. This review is a great compilation of all the data published in the field of Host-pathogen response against Toxoplasma gondii. However, the presentation of data in some places is a little dumped. I offered some major corrections in the manuscript.

·      Overall, English needs to improve.

·      I have reviewed some citations and found that the author citing the paper does not correspond to the claimed data.

·      At 47-line author claimed regarding the sexual stage in mice depends on the availability of linoleic acid. Need to rewrite it again.

·      At 100-line regarding infection of Toxoplasma, the Author should add undercooked meat

·      At 116-line Author mentioned “DCs to natural killer (NK)’ giving citation 47, but in this paper (47) they do not mention NK cells and they mentioned only DCs trigger more parasite dissemination, please cite the appropriate paper here.

·      At 328-line, citation 123 doesn’t directly confirm the HSP65, please cite the appropriate paper here

·      Line 445 to 447, please change the wording. It’s very confusing

·      Line 461 to 491, Its literally so much data added here. Need to explain properly in this paragraph. Try to connect the story, as so much data that it’s hard to get what the author wants to explain.

·       Line 515 and 516, Please rewrite this sentence as it repetitions.

Overall, I feel data is added at times with little discussion. Need to explain properly. 

Author Response

On behalf of my co-authors, I thank you very much for giving us an opportunity to revise our manuscript, we appreciate very much for the editor’s and reviewer’s positive and constructive comments and suggestions on our manuscript entitled " Overview of apoptosis, autophagy, and inflammatory processes in Toxoplasma gondii infected cells". We have studied reviewers’ comments carefully and tried our best to revise our manuscript according to them. All suggestions were separately considered, addressed point-by-point and all of edited changes were highlighted. We hope that the revised form of our manuscript meets the “Pathogens” journal and would be satisfactory for acceptance.

Answers to the comments

Reviewer 3

This is a straightforward review reporting overview of the immune response against Toxoplasma gondii. This review is a great compilation of all the data published in the field of Host-pathogen response against Toxoplasma gondii. However, the presentation of data in some places is a little dumped. I offered some major corrections in the manuscript.

We would like to thank you very much for the very thoughtful review that helped us greatly to revise the manuscript and to improve its quality. We hope that the revised form of our manuscript meets the respect reviewer and would be satisfactory for acceptance.

Comment: Overall, English needs to improve.

Answer: According to your constructive comment, the manuscript has been studied several times by the authors, and its grammatical and structural problems were corrected. In addition, the manuscript was completely revised by English editors and all changes were highlighted (green) in the manuscript.

Comment: I have reviewed some citations and found that the author citing the paper does not correspond to the claimed data.

Answer: Thank you for constructive comment. We carefully checked all the references in the text. Some references that did not match the contents were removed.

-List of deleted citations:

  1. Elmore, S.A.; Jones, J.L.; Conrad, P.A.; Patton, S.; Lindsay, D.S.; Dubey, J. Toxoplasma gondii: epidemiology, feline clinical aspects, and prevention. Trends in parasitology 2010, 26, 190-196.
  2. Howe, D.K.; Honoré, S.; Derouin, F.; Sibley, L.D. Determination of genotypes of Toxoplasma gondii strains isolated from patients with toxoplasmosis. Journal of clinical microbiology 1997, 35, 1411-1414.
  3. Frenkel, J. Host, strain and treatment variation as factors in the pathogenesis of toxoplasmosis. The American journal of tropical medicine and hygiene 1953, 2, 390-415.

33: Alvarado-Esquivel, C.; Alanis-Quiñones, O.-P.; Arreola-Valenzuela, M.-Á.; Rodríguez-Briones, A.; Piedra-Nevarez, L.-J.; Duran-Morales, E.; Estrada-Martínez, S.; Martínez-García, S.-A.; Liesenfeld, O. Seroepidemiology of Toxoplasma gondii infection in psychiatric inpatients in a northern Mexican city. BMC Infectious Diseases 2006, 6, 178.

  1. Ben-Baruch, A. Host microenvironment in breast cancer development: inflammatory cells, cytokines and chemokines in breast cancer progression: reciprocal tumor–microenvironment interactions. Breast cancer research 2002, 5, 31.
  2. Tan, B.H.; Meinken, C.; Bastian, M.; Bruns, H.; Legaspi, A.; Ochoa, M.T.; Krutzik, S.R.; Bloom, B.R.; Ganz, T.; Modlin, R.L. Macrophages acquire neutrophil granules for antimicrobial activity against intracellular pathogens. The Journal of Immunology 2006, 177, 1864-1871.
  3. Lämmermann, T.; Afonso, P.V.; Angermann, B.R.; Wang, J.M.; Kastenmüller, W.; Parent, C.A.; Germain, R.N. Neutrophil swarms require LTB4 and integrins at sites of cell death in vivo. Nature 2013, 498, 371.
  4. Rahimi, H.M.; Nemati, S.; Alavifard, H.; Baghaei, K.; Mirjalali, H.; Zali, M.R. Soluble total antigen derived from Toxoplasma gondii RH strain prevents apoptosis, but induces anti-apoptosis in human monocyte cell line. Folia Parasitologica 2021, 68, 1-7.

174.Tan, X.; Thapa, N.; Sun, Y.; Anderson, R.A. A kinase-independent role for EGF receptor in autophagy initiation. Cell 2015, 160, 145-160.

  1. Neumann, A.K.; Yang, J.; Biju, M.P.; Joseph, S.K.; Johnson, R.S.; Haase, V.H.; Freedman, B.D.; Turka, L.A. Hypoxia inducible factor 1α regulates T cell receptor signal transduction. Proceedings of the National Academy of Sciences 2005, 102, 17071-17076.

- According to the corrections in the manuscript, new references were added to the text as follows:

  1. Lambert, H.; Barragan, A. Modelling parasite dissemination: host cell subversion and immune evasion by Toxoplasma gondii. Cellular microbiology 2010, 12, 292-300.
  2. Dubey, J.; Murata, F.; Cerqueira-Cézar, C.; Kwok, O. Epidemiologic and public health significance of Toxoplasma gondii infections in venison: 2009–2020. The Journal of Parasitology 2021, 107, 309-319.
  3. Ahmadpour, E.; Rahimi, M.T.; Ghojoghi, A.; Rezaei, F.; Hatam-Nahavandi, K.; Oliveira, S.M.; de Lourdes Pereira, M.; Majidiani, H.; Siyadatpanah, A.; Elhamirad, S. Toxoplasma gondii Infection in Marine Animal Species, as a Potential Source of Food Contamination: A Systematic Review and Meta-Analysis. Acta parasitologica 2022, 1-14.
  4. Sultana, M.A.; Du, A.; Carow, B.; Angbjär, C.M.; Weidner, J.M.; Kanatani, S.; Fuks, J.M.; Muliaditan, T.; James, J.; Mansfield, I.O. Downmodulation of effector functions in NK cells upon Toxoplasma gondii infection. Infection and immunity 2017, 85, e00069-00017.
  5. Nash, P.B.; Purner, M.B.; Leon, R.P.; Clarke, P.; Duke, R.C.; Curiel, T.J. Toxoplasma gondii-infected cells are resistant to multiple inducers of apoptosis. The Journal of Immunology 1998, 160, 1824-1830.
  6. Hisaeda, H.; Sakai, T.; Ishikawa, H.; Maekawa, Y.; Yasutomo, K.; Good, R.A.; Himeno, K. Heat shock protein 65 induced by gammadelta T cells prevents apoptosis of macrophages and contributes to host defense in mice infected with Toxoplasma gondii. The Journal of Immunology 1997, 159, 2375-2381.
  7. Blader, I.J.; Koshy, A.A. Toxoplasma Development of Its Replicative Niche: In Its Host Cell and Beyond. Eukaryotic cell 2014, EC. 00081-00014.

At 47-line author claimed regarding the sexual stage in mice depends on the availability of linoleic acid. Need to rewrite it again.

Answer: Thanks for your attention, it was corrected and highlighted in the Text, as follow:

"New research shows that we can have sexual T. gondii in mice. Felines are the only mammals that lack delta-6-desaturase activity in their intestines, which is required for linoleic acid metabolism, resulting in systemic excess of linoleic acid. It was found that inhibition of murine delta-6-desaturase and supplementation of their diet with linoleic acid allowed T. gondii sexual development in mice" (Page 2, lines 48-52).

At 100-line regarding infection of Toxoplasma, the Author should add undercooked meat

Answer: It was added in the Text (Page 3, line 101).

At 116-line Author mentioned “DCs to natural killer (NK)’ giving citation 47, but in this paper (47) they do not mention NK cells and they mentioned only DCs trigger more parasite dissemination, please cite the appropriate paper here.

Answer: Thanks for your attention, new related reference was added and highlighted.

  1. Persson, C.M.; Lambert, H.; Vutova, P.P.; Dellacasa-Lindberg, I.; Nederby, J.; Yagita, H.; Ljunggren, H.-G.; Grandien, A.; Barragan, A.; Chambers, B.J. Transmission of Toxoplasma gondii from infected dendritic cells to natural killer cells. Infection and immunity 2009, 77, 970-976.
  2. Sultana, M.A.; Du, A.; Carow, B.; Angbjär, C.M.; Weidner, J.M.; Kanatani, S.; Fuks, J.M.; Muliaditan, T.; James, J.; Mansfield, I.O. Downmodulation of effector functions in NK cells upon Toxoplasma gondii infection. Infection and immunity 2017, 85, e00069-00017.

At 328-line, citation 123 doesn’t directly confirm the HSP65, please cite the appropriate paper here

Answer: Thanks for your attention, it was performed.

  1. Hisaeda, H.; Sakai, T.; Ishikawa, H.; Maekawa, Y.; Yasutomo, K.; Good, R.A.; Himeno, K. Heat shock protein 65 induced by gammadelta T cells prevents apoptosis of macrophages and contributes to host defense in mice infected with Toxoplasma gondii. The Journal of Immunology 1997, 159, 2375-2381.

Line 445 to 447, please change the wording. It’s very confusing

Answer: It was performed and highlighted in the Text, as follow:

How exactly Toxoplasma is restricted in a human cell might depend on the phagocytic ability of the cell versus induced GBP and IDO levels. Surprisingly, GBP versus IDO-mediated restriction of Toxoplasma has not been studied. It is thus possible that these pathways counter-regulate each other and co-exist (page 11, lines 470-473).

Line 461 to 491, its literally so much data added here. Need to explain properly in this paragraph. Try to connect the story, as so much data that it’s hard to get what the author wants to explain.

Answer: Thank you for constructive comment. This paragraph shows the cascade of autophagy processes and corrections were made as far as possible (pages 11-12, line 486-515).

Line 515 and 516, Please rewrite this sentence as it repetitions.

Answer: It was performed and highlighted in the Text, as follow:

Studies have shown that tyrosine kinase inhibitors and EGFR can inhibit the phagolysosomes in parasites. EGFR activation depends on the expression of at least two micronimeric proteins, MIC1 and MIC3. T. gondii micronemal proteins containing EGF domains appeared to promote EGFR activation (page 12, lines 537-540)

Overall, I feel data is added at times with little discussion. Need to explain properly. 

Answer: Thank you for constructive comment. Our research team carefully reviewed the entire manuscript. The sentences were completed in some paragraphs (blue color), as follow: (page 3, lines 112-116), (page 4, lines 163-164), (page 4, lines 182-184), (page 5, lines 189-193), (page 5, lines 219-223), (page 5, lines 231-235), (page 6-7, lines 269-272), (page 7, lines 293-296), (page 7, lines 316-319), (page 8, lines 325-327), (page 8, lines 361-364), (page 9, lines 383-387), (page 13, lines 566-570), (page 13, lines 586-588), (page 14, lines 644-646), and (page 15, lines 687-691).

I deeply appreciate your kind favor and cooperation.

With the best regards, sincerely yours

Abdol Sattar Pagheh, Ph.D

Round 2

Reviewer 3 Report (New Reviewer)

The authors have addressed the main concerns of the article. Therefore, I support the publication of this work.

This manuscript is a resubmission of an earlier submission. The following is a list of the peer review reports and author responses from that submission.

Round 1

Reviewer 1 Report

Ahmadpour et al. discussed the apoptosis, autophagy, and inflammatory processes in Toxoplasma gondii-infected cells.  The review is very interesting thus a better comprehension of the cellular processes by which T. gondii immunoregulation host cell death pathways will boost our conception of the host-parasite interplay in toxoplasmosis. I suggest the publication of this nice manuscript, however, please check the right use of abbreviations throughout the manuscript. Also, T. gondii should be italic throughout the manuscript. Please see the attached file for minor comments that might be considered.

Author Response

Dear Editor-in-Chief:                                                                         

On behalf of my co-authors, I thank you very much for giving us an opportunity to revise our manuscript, we appreciate very much for the editor’s and reviewer’s positive and constructive comments and suggestions on our manuscript entitled " Overview of apoptosis, autophagy, and inflammatory processes in Toxoplasma gondii infected cells". We have studied reviewers’ comments carefully and tried our best to revise our manuscript according to them. All suggestions were separately considered, addressed point-by-point and all of edited changes were

+highlighted. We hope that the revised form of our manuscript meets the “Pathogens” journal and would be satisfactory for acceptance.

Answers to comments of Reviewers

Reviewer 1:

Ahmadpour et al. discussed the apoptosis, autophagy, and inflammatory processes in Toxoplasma gondii-infected cells.  The review is very interesting thus a better comprehension of the cellular processes by which T. gondii immunoregulation host cell death pathways will boost our conception of the host-parasite interplay in toxoplasmosis. I suggest the publication of this nice manuscript, however, please check the right use of abbreviations throughout the manuscript. Also, T. gondii should be italic throughout the manuscript. Please see the attached file for minor comments that might be considered.

Answer: We would like to thank you very much for the very thoughtful review that helped us greatly to revise the manuscript and to improve its quality. In accordance with the comments of the respected reviewer, the mentioned items were corrected.

I deeply appreciate your kind favor and cooperation.

With the best regards, sincerely yours

Abdol Sattar Pagheh, Ph.D.

Reviewer 2 Report

In this review, Ahmadpour et al. summarize the roles of regulated cell death and various other immune responses during Toxoplasma gondii (T. gondii) infection. The review is exhaustive and I can see the effort put into this manuscript.

However, there are massive flaws concerning the English writing, phrasing and the grammar. For example complete sentences, enumerations and passages are placed in highly inappropriate places, which do not fit thematically at all. Some text passages, on the other side, are of high quality and well placed, which clearly shows that authors with high differences in style and skill of scientific writing worked on this manuscript. This leads to an manuscript draft, which makes the impression of being badly crafted together.

One or two rounds more of careful corrective reading of the final draft of the manuscript before sending it to this journal would have erased most of the errors and inappropriately placed passages. Not every scientist is a native English speaker, but again many of the errors could have been corrected with just careful reading of the final manuscript.

Most critically, the figures in the article only depict cell death pathways in general. Just putting an arrow between a T. gondii parasite and the cell death pathway is not enough to display and explain the connection of the parasite and the cell death pathway. The general components of the cell death pathways can be seen in any other general review concerning cell death. But in a review about T. gondii, figures, which depict the interplay of the parasite with the cellular pathways within the host cells, are highly needed. However, these are completely missing in this article. Moreover, all three figures depict only cell death pathways, but other cellular processes (autophagy, STAT signaling), which can be influenced and targeted by T. gondii are not depicted in the manuscript at all. Similarly a figure, which, clarifies the lifecycle and main infection routes, as well as the three virulence stages would have been much more useful than three figures which only describe cell death pathways in general. All three figures have no impact for the understanding of this article and could be completely removed without any disadvantage for the manuscript.

This article despite the fact that effort was put into it does not meet the quality standards of Pathogens. Because of the statements made above and since the flaws of this article cannot be corrected with a major revision I have to reject the article.

Author Response

Dear Editor-in-Chief:                                                                         

On behalf of my co-authors, I thank you very much for giving us an opportunity to revise our manuscript, we appreciate very much for the editor’s and reviewer’s positive and constructive comments and suggestions on our manuscript entitled " Overview of apoptosis, autophagy, and inflammatory processes in Toxoplasma gondii infected cells". We have studied reviewers’ comments carefully and tried our best to revise our manuscript according to them. All suggestions were separately considered, addressed point-by-point and all of edited changes were highlighted. We hope that the revised form of our manuscript meets the “Pathogens” journal and would be satisfactory for acceptance.

Answers to comments of Reviewers

Reviewer 2:

In this review, Ahmadpour et al. summarize the roles of regulated cell death and various other immune responses during Toxoplasma gondii (T. gondii) infection. The review is exhaustive and I can see the effort put into this manuscript.

However, there are massive flaws concerning the English writing, phrasing and the grammar. For example complete sentences, enumerations and passages are placed in highly inappropriate places, which do not fit thematically at all. Some text passages, on the other side, are of high quality and well placed, which clearly shows that authors with high differences in style and skill of scientific writing worked on this manuscript. This leads to an manuscript draft, which makes the impression of being badly crafted together.

One or two rounds more of careful corrective reading of the final draft of the manuscript before sending it to this journal would have erased most of the errors and inappropriately placed passages. Not every scientist is a native English speaker, but again many of the errors could have been corrected with just careful reading of the final manuscript.

Most critically, the figures in the article only depict cell death pathways in general. Just putting an arrow between a T. gondii parasite and the cell death pathway is not enough to display and explain the connection of the parasite and the cell death pathway. The general components of the cell death pathways can be seen in any other general review concerning cell death. But in a review about T. gondii, figures, which depict the interplay of the parasite with the cellular pathways within the host cells, are highly needed. However, these are completely missing in this article. Moreover, all three figures depict only cell death pathways, but other cellular processes (autophagy, STAT signaling), which can be influenced and targeted by T. gondii are not depicted in the manuscript at all. Similarly a figure, which, clarifies the lifecycle and main infection routes, as well as the three virulence stages would have been much more useful than three figures which only describe cell death pathways in general. All three figures have no impact for the understanding of this article and could be completely removed without any disadvantage for the manuscript.

This article despite the fact that effort was put into it does not meet the quality standards of Pathogens. Because of the statements made above and since the flaws of this article cannot be corrected with a major revision I have to reject the article.

Answer: We would like to thank you very much for the very thoughtful review that helped us greatly to revise the manuscript and to improve its quality. In accordance with the comments of respected reviewer, the manuscript was completely revised by a native English editor and all changes were highlighted in the manuscript.

Regarding Figuesd 1-3, it should be mentioned that according to the format of review articles, our aim was to summarize the described pathways and show important factors. Therefore, we think that if all the factors related to cell death and Toxoplasma infection were added in the figures, it would be confusing and incomprehensible. It is necessary to mention that we first designed all three figures in the following form:

But as mentioned, we drew them separately for better understanding of the contents. Also, regarding the life cycle of the parasite, due to the existence of complete and comprehensive forms in previous studies, we omitted it.

I deeply appreciate your kind favor and cooperation.

With the best regards, sincerely yours

Abdol Sattar Pagheh, Ph.D

Reviewer 3 Report

Thank you for this very good overview on T. gondii pathogenesis aspects.

In a review, we need to make some choices because of the broad of the subject. You focused more on specific immunity than on innate immunity but I accept this point of view.

I have a couple of small remarks

L86: Type III strains are considered as avirulent (instead of a virulent???).

L122: (ROPS) and (space)

L123: (GRAs) can  (space)

L241 to L253: Is apoptosis a normal process to eliminate cells infected by intracellular parasites? If yes, maybe write this kind of sentence at top of the paragraph.

For the interaction between pyroptosis and T. gondii, the article of Wang et al. 2020 in frontiers in Immunology could be cited.

Author Response

Dear Editor-in-Chief:                                                                          

On behalf of my co-authors, I thank you very much for giving us an opportunity to revise our manuscript, we appreciate very much for the editor’s and reviewer’s positive and constructive comments and suggestions on our manuscript entitled " Overview of apoptosis, autophagy, and inflammatory processes in Toxoplasma gondii infected cells". We have studied reviewers’ comments carefully and tried our best to revise our manuscript according to them. All suggestions were separately considered, addressed point-by-point and all of edited changes were highlighted. We hope that the revised form of our manuscript meets the “Pathogens” journal and would be satisfactory for acceptance.

Answers to comments of Reviewers

Reviewer 3:

Thank you for this very good overview on T. gondii pathogenesis aspects. In a review, we need to make some choices because of the broad of the subject. You focused more on specific immunity than on innate immunity but I accept this point of view.

 Answer: We would like to thank you very much for the very thoughtful review that helped us greatly to revise the manuscript and to improve its quality.

I have a couple of small remarks

 L86: Type III strains are considered as avirulent (instead of a virulent???).

 Answer: Thank you for your attention. It was corrected and highlighted.

L122: (ROPS) and (space)

 Answer: Thank you for your attention. It was corrected

 L123: (GRAs) can (space)

  Answer: Thank you for your attention. It was corrected

L241 to L253: Is apoptosis a normal process to eliminate cells infected by intracellular parasites? If yes, maybe write this kind of sentence at top of the paragraph.

Answer: Thank you for your suggestion. The question of whether apoptosis performs a physiologic function during Toxoplasma infection is key to understanding host–parasite interactions. It is mentioned in the text that “evidence suggests that T. gondii may inhibit caspase activity, triggering probiotic, and anti-apoptotic responses in host cells”, but it has not been proven yet. Indeed, T. gondii can use strategies to manipulate the process of cell death that can accelerate or delay apoptosis in favor of the parasite. Therefore, we need greater knowledge about how the parasite modulates host cell death.

For the interaction between pyroptosis and T. gondii, the article of Wang et al. 2020 in frontiers in Immunology could be cited.

Answer: In accordance with the comments of the respected reviewer, it was added and highlighted.

I deeply appreciate your kind favor and cooperation.

With the best regards, sincerely yours

Abdol Sattar Pagheh, Ph.D.

Round 2

Reviewer 2 Report

The authors again put much effort in the corrections of the text. The figures are still not appropiate (despite the fact that they are correctly depicted in terms of cell death), but the text alone might help some researchers to get an overview of this topic. Therefore, it would be not fair to put that much work into the drawer. I accept the article, but I can only highly encourage the authors for puttting much more effort also in the figures in future articles.
